# Rational design of DNA nanostructures for single molecule biosensing

Mukhil Raveendran[1], Andrew J. Lee [1,2], Rajan Sharma[1], Christoph Wälti[1,2] & Paolo Actis [1,2 ✉]

The ability to detect low concentrations of biomarkers in patient samples is one of the cornerstones of modern healthcare. In general, biosensing approaches are based on measuring signals resulting from the interaction of a large ensemble of molecules with the sensor. Here, we report a biosensor platform using DNA origami featuring a central cavity with a target-specific DNA aptamer coupled with a nanopore read-out to enable individual biomarker detection. We show that the modulation of the ion current through the nanopore upon the DNA origami translocation strongly depends on the presence of the biomarker in the cavity. We exploit this to generate a biosensing platform with a limit of detection of 3 nM and capable of the detection of human C-reactive protein (CRP) in clinically relevant fluids. Future development of this approach may enable multiplexed biomarker detection by using ribbons of DNA origami with integrated barcoding.

[1] School of Electronic and Electrical Engineering and Pollard Institute, University of Leeds, Leeds, UK. [2] Bragg Centre for Materials Research, University of Leeds, Leeds, UK. ✉email: p.actis@leeds.ac.uk

Rapid and low-cost detection of disease biomarkers is becoming increasingly important in modern healthcare where a growing focus is placed on early diagnosis[1–4]. This presents considerable technological challenges as the relevant biomarkers are often present at very small concentrations. Ideally, a diagnostic assay should be able to detect the presence of few biomarkers in a small volume of a complex clinical sample. The currently employed clinical diagnostic assays are generally based on ensemble-averaging immunoassays such as enzyme-linked immunosorbent assays (ELISAs). In these, an antibody or antibody mimetic is used to capture relevant biomarkers in the sample, and in general, each antibody–biomarker interaction contributes a small amount to the accumulated assay signal. However, the individual immuno-interactions cannot be identified anymore and only manifest themselves as part of this ensemble-averaged signal. Arguably, the ability to detect biomarkers with single entity resolution rather than via ensemble-averaging techniques provides significant advantages for the detection of ultra-small biomarker concentrations.

Of the many single molecule methods which have been developed over recent years[5–10], the use of nanopores, where a voltage is applied across the nanopore and pulses in the time-varying electrochemical current are used for detecting individual proteins[11–16], is a promising approach. Here, single molecules translocate through the nanoscale pore causing a momentary modulation in the otherwise steady ion current[17]. This approach has been employed in a number of single-molecule protein and DNA detection studies[18–22].

However, despite these advances, there remain significant challenges. It is very difficult to detect a specific molecule from a complex mixture as the translocation speed of single entities is generally high when the diameter of the nanopore is much larger than the size of the molecule, and the translocation therefore results in only a weak signal, and correspondingly low signal-to-noise. Furthermore, the signals resulting from different proteins are generally very similar and difficult to differentiate. To address these limitations, a variety of strategies have been implemented, including slowing down protein translocations through careful selection of electrolytes[23,24] and developing high-bandwidth electronics[25,26]. Also, chemical and biological modification of nanopores[27,28] and the use of hybrid nanopores[29] have been explored to increase sensitivity and selectivity. More recently, large carrier molecules including nanoparticles[11], antibodies[13] and very promisingly long linear dsDNA[29–33] and the entire genome of the lambda bacteriophage[14] have been exploited both to host the protein of interest and to reduce the translocation speed.

While DNA-based carriers have been used successfully and indeed provide a useful carrier system owing to the ease of modifying and engineering the system, the long linear DNA carriers are not without limitations – they are prone to forming knots and kinks[34] which are known to lead to false positive signals during nanopore translocation. Furthermore, significant variations in the signal have been noted, depending on the translocation and orientation, as well as high blockage rates due to the simultaneous passage of multiple DNA molecules through the nanopore[35,36]. In an approach to circumvent these problems, recent work by Cai et al.[32] combined single-molecule fluorescence with nanopore detection to perform single-molecule binding assays. Their approach, although elegant, negates the key advantage of nanopore sensing by introducing labelling and complex optical setups as opposed to a simple electrical label-free read-out.

An alternative approach to long linear DNA carriers is to form a carrier molecule by folding up a long DNA into a defined robust geometry which can then host the subject molecule, in an analogous approach to that used to study single biological reactions with high speed atomic force microscopy[37]. These large robust DNA nanostructures, referred to as DNA origami[38], provide a versatile carrier platform to circumvent the limitations of linear DNA-based approaches. Importantly, our previous work has demonstrated that single 2D DNA origami can be detected using nanopores and that their translocations yield different ion current signatures depending on their geometries[39].

Here we exploit rationally designed 2D DNA origami nanostructures for quantitative single-molecule biosensing. We demonstrate that DNA aptamer-functionalized DNA origami can capture the analyte of interest and that the occupied and unoccupied DNA origami's translocation fingerprints are readily distinguishable. We demonstrate that in addition to the characteristically different peak shape, also the peak amplitude and dwell time can be used to distinguish occupied from unoccupied carriers. Taken together, this enables quantitative biosensing via counting of individual occupied DNA origami carriers, which we demonstrate in physiological solutions and diluted human plasma.

## Results

**Concentric squares**. In a previous study we showed that the ion current fingerprint of 2D DNA origami upon translocation through a nanopore depends on their geometries[39]. In particular, we found that the presence of a cavity in the centre of the origami led to the splitting of the peak in the ion current into a double peak. Here, we hypothesise that if a DNA origami is designed to contain a cavity in the centre into which an analyte of interest can bind specifically, the presence or absence of the analyte from the origami can be detected using a nanopore by observing the characteristic translocation ion current peak. The frequencies of the two characteristically different peaks can be computed from the observation of a large number of individual peaks to obtain an analyte concentration-dependent signal.

To gain further insight into the design parameters of the cavity in the 2D DNA origami governing the transition from single to double-peak, we designed three different DNA nanostructures with identical outer but variable cavity dimensions. The three DNA nanostructures resemble a set of three concentric squares and are referred to as ConA for a solid nanostructure (100 nm × 85 nm), ConB and ConC for a nanostructure of identical outer dimension but containing a central cavity of 30 nm × 12 nm and 65 nm × 25 nm, respectively, as shown in Fig. 1a.

The concentric square nanostructures were all designed to be folded according to established DNA origami principles. The three DNA tiles were made from the same constituents, i.e., the same oligonucleotide staples and identical scaffold routing, with the varying cavity dimensions achieved by shortening the underlying DNA scaffold. For example, the scaffold DNA used to assemble ConB was the same as the one used for assembling ConA but with the section that folds the central part removed. This approach ensures that as much of the structure as possible remained identical between the different DNA origamis while varying the cavity dimension so we can directly correlate the ion current signature to the cavity volume. The detailed procedure for custom scaffold design and production as well as the routing map are provided in Supplementary information (Supplementary Fig. 11a–d).

The folded DNA nanostructures were imaged with Atomic Force Microscopy (AFM) to confirm successful assembly. Representative AFM images are shown in Fig. 1b and it can be seen that all three structures were formed as intended. Not unexpectedly, the measured dimensions of the three tiles (all averaged over 30 tiles) differ slightly from the designs. The

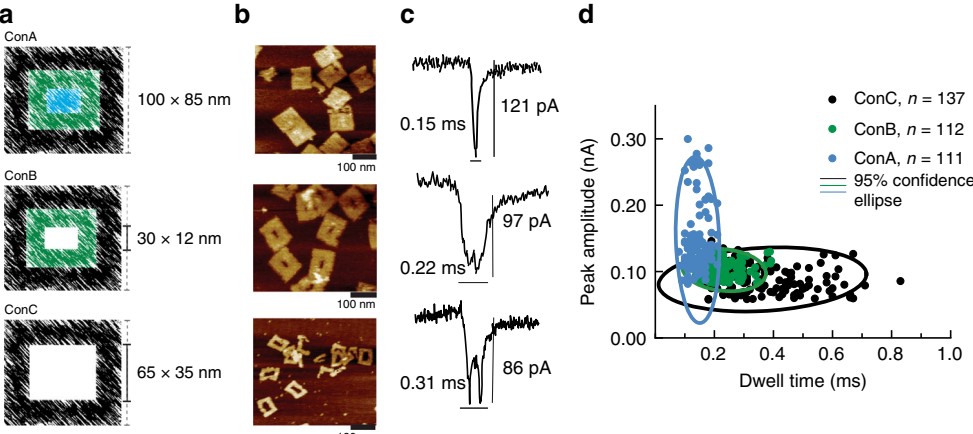

**Fig. 1 Concentric square DNA origami nanostructures. a** Colour-coded schematic representation of DNA nanostructure designs – ConA is a solid tile and ConB and ConC are frame-like with different central cavities. All three structures feature similar external dimensions but varying internal cavity. **b** AFM micrographs of the respective nanostructures and **c** their typical translocation ion current signatures. **d** Scatter plot of individual translocation events with peak amplitude plotted versus dwell time for ConA (blue circles), ConB (green circles) and ConC (black circles), overlaid with the respective 95% confidence ellipses. Source data are in the Source Data file.

outer dimensions of all tiles were found to be 107 nm × 86 nm, and ConB and ConC with a cavity of 29 nm × 11 nm and 65 nm × 38 nm, respectively.

To carry out nanopore translocation studies for the concentric square samples, we employed glass nanopipettes with ~100 nm pore diameter (Supplementary Fig. 10a, b). The nanopipettes were fabricated via laser pulling and showed a resistance of 86 ± 10 MΩ in 0.1 M KCl. Translocation experiments were carried out by loading a sample solution at a concentration of 500 pM into the nanopipette and recording the ion current under a voltage of −350 mV.

Figure 1c shows representative ion current signatures for the different nanostructures translocating through the nanopore. As expected, as a cavity is introduced in the nanostructure the observed peak structure changes from a single peak (no cavity present) to a double peak (cavity present), with the double peak becoming more pronounced (double peak height of 37 pA for ConC vs 14 pA for ConB, Supplementary Fig. 2) with increasing cavity size. A detailed quantitative analysis of more than 100 ion current peaks for each DNA nanostructure shows that the average peak amplitude and dwell time differ significantly for different nanostructures (Supplementary Fig. 1).

ConC (large cavity) translocates through the nanopore significantly slower (dwell time = 0.31 ± 0.14 ms) and leads to a smaller ion current increase (peak amplitude = 86 ± 19 pA) when compared to ConB (small cavity; dwell time = 0.22 ± 0.06 ms and peak amplitude = 97 ± 13 pA). In contrast, ConA (no cavity) nanostructure causes the largest ion current increase (peak amplitude = 121 ± 23 pA) with the shortest dwell time (0.15 ± 0.01 ms). It is possible that these differences in peak amplitude and dwell time could represent the combined contribution of the overall charge and geometry of the DNA nanostructures due to the variable scaffold length used in these designs. It is, however, notable that the distinct ion current signatures consistently correlate with the cavity geometry, both in this study as well as our previous study on different tiles with cavities[39]. When taken together, these parameters allow the identification of distinct populations of DNA nanostructures (Fig. 1d).

**DNA nanostructures as translocation carriers in biosensing**. The relationship of ion current peak signature to cavity geometry within these DNA nanostructures provides a unique opportunity to detect the presence of a much smaller molecule.

If a binding moiety specific to the target molecule of interest is placed within the cavity of the DNA nanostructure, the binding of a target molecule to the binding moiety partially fills the cavity which in turn results in a characteristic change to the ion current signature (Fig. 2a).

As proof of principle, we demonstrate the quantitative detection of human C-reactive protein (CRP), which is an established inflammation biomarker. In a healthy adult the median CRP concentration is 0.8 µg/mL, and its concentration in blood exceed 1 mg/mL (8 µM)[40] as a result of an inflammatory response. CRP exists as a pentamer with a molecular weight of ~125 kDa and a size of ~11 nm (Supplementary Fig. 5). We designed a DNA nanostructure with a central cavity large enough for a CRP molecule to fit in but at the same time small enough such that the presence of CRP in the cavity will make a measurable difference to the translocation ion current. As informed by the study with concentric squares, the nanostructure was designed with a central cavity of 35 nm × 35 nm to provide a robust double peak signature (Fig. 2b).

An internal anchor stub was introduced within the central cavity such that a specific capture moiety can be placed inside the cavity via DNA hybridisation (Fig. 2b). In order to enable selective CRP detection, we chose two well-characterised CRP-DNA aptamers (aptamer 1 and aptamer 2) from the literature[41–43] as potential capture moieties. We note that while the sequences of the DNA aptamers employed here are the same as the ones published, for this application the aptamers have to be extended at their 5′ or 3′ end, respectively, so they can be hybridised into the cavity. To ensure the DNA aptamers are still performing as intended, we confirmed CRP binding to the extended aptamers by surface plasmon resonance (SPR). As shown in Supplementary Fig. 3, both DNA aptamers continue to bind CRP as expected. Aptamer 1 showed faster binding kinetics compared to aptamer 2 and hence was selected for the detailed demonstration of the biosensor concept.

A notch was also included at one corner of the DNA origami to act as a polarity marker to identify the orientation of the origami in AFM images (Fig. 2b). The complete DNA nanostructure and aptamer assembly is henceforth referred to as the carrier.

The performance of the DNA aptamer within the context of the carrier (i.e. aptamer hybridised within the DNA nanostructure, Fig. 2b) was investigated by gel retardation assay. A 9 nM solution of the carrier was incubated with increasing

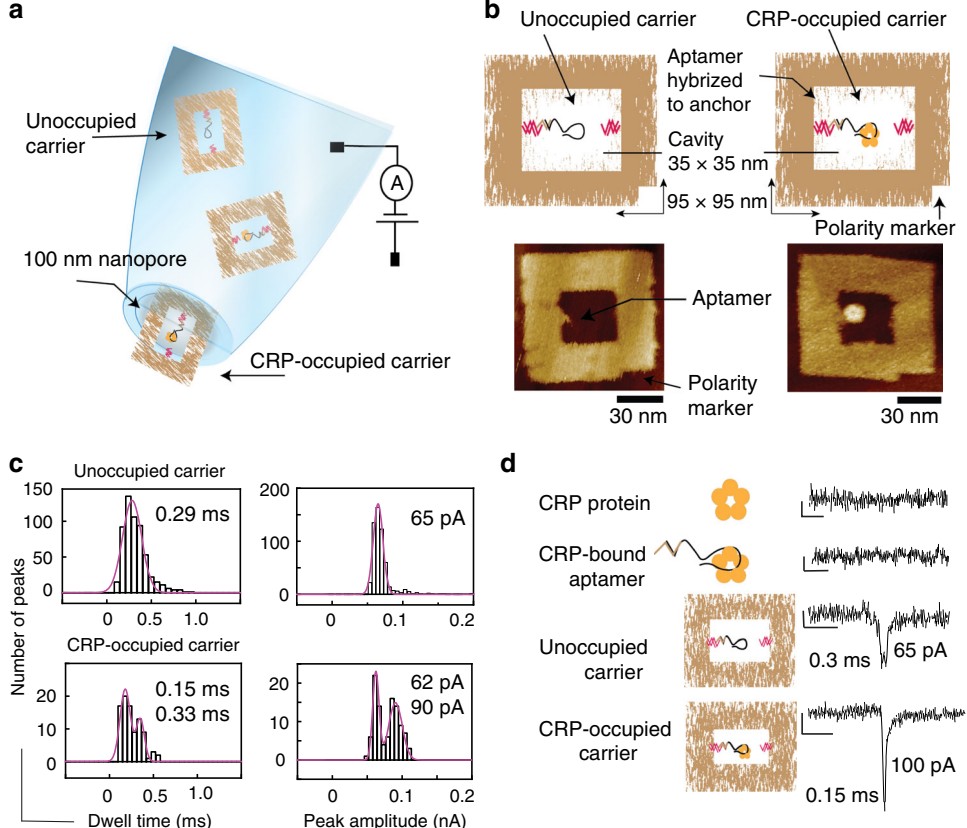

**Fig. 2 DNA origami carrier-based biosensors. a** Schematic representation of the DNA nanostructure-based biosensing concept exploiting translocation through nanopipettes as the sensing mechanism. **b** Schematic representation of the design and representative AFM micrographs of the unoccupied and occupied DNA origami carriers. The frame DNA nanostructure is ~95 nm × 95 nm in dimension with a 35 nm × 35 nm inner cavity. The DNA origami comprises small nucleotide 'anchors' that protrude into the cavity which facilitate the incorporation of the DNA aptamer via hybridisation. The DNA carrier also includes a polarity marker. **c** Peak amplitude and dwell time histograms of DNA nanostructure carriers (9 nM) and carriers incubated with CRP at 9 nM concentration, both measured at a final carrier concentration of 500 pM. **d** Typical ion current signatures upon translocation of CRP molecules, CRP–DNA aptamer complexes, unoccupied carriers and CRP-bound occupied carriers. The scale bars represent 20 pA and 1 ms, respectively. Source data are in the Source Data file.

concentrations of CRP (9, 18, 27 and 36 nM) in the translocation buffer (0.1 M KCl containing 10 mM MgAc, 2 mM CaCl$_2$, 10 mM TrisAc and 1 mM EDTA) at room temperature for 30 min. The gel is shown in Fig. S4, and a concentration-dependent shift of the carrier bands can be seen which demonstrates successful CRP binding to the carrier. CRP (36 nM) binding to the carrier (9 nM) was further confirmed by AFM under similar buffer conditions as stated above (Fig. 2b), where the CPR is clearly observed in the cavity bound to the aptamer which is opposite the inbuilt polarity marker.

To exploit the DNA nanostructure carrier in single-molecule nanopore sensing, the occupied and unoccupied carriers need to have unique fingerprints (composed of dwell time, amplitude and shape of the ion current peak) which relate to the presence of CRP within the carrier. Nanopore translocation studies of the carriers (at 500 pM concentration) using glass nanopipettes were carried out as above. The quantitative analysis of more than 100 translocation peaks reveals a peak amplitude of 65 ± 6 pA and dwell time of 0.29 ± 0.1 ms (Fig. 2c, top) which are comparable with previous observations of similar-sized DNA nanostructures containing cavities[39].

Similarly, to analyse the CRP-occupied carriers we used an equimolar solution of CRP and carriers at 9 nM with the aim of having a mixed population of occupied and unoccupied carriers. The solution was incubated at room temperature for 30 min in

the translocation buffer and subjected to nanopipette translocation at 500 pM concentration, which led to a mixed population of single and double peaks in the ion current. The analysis of the peak amplitudes and dwell times of the two classes of peaks revealed a bipolar distribution. The double peaks had an average peak amplitude of 62 ± 4 pA and dwell time of 0.33 ± 0.03 ms (Fig. 2c, bottom; n > 100). In contrast, the analysis of more than 50 single peaks revealed an average dwell time and peak amplitude of 0.15 ± 0.05 ms and 90 ± 9 pA, respectively, which demonstrates a substantial shift in both quantities as a result of CRP binding (Fig. 2c, bottom). We note that the lower peak amplitude and the longer dwell time are very similar to the ones observed for the unoccupied carriers.

Together with the results of the concentric squares study which showed that the peak shape changes from a double peak to a single peak upon filling in the central cavity, we speculate that the single peak events correspond to occupied carriers, i.e., carriers with a CRP bound to the specific aptamer. Furthermore, the double peaks are accounting for ~30% of the total number of observed events. This is in line with the percentage of occupied carriers that would be expected based on the $K_d$ obtained from the SPR experiments (Supplementary Fig. 3d), which suggests that the percentage is concentration dependent as expected.

However, in order to use this approach for high-sensitivity biosensing, a reliable way of classifying the different ion current

events is required. Supplementary Fig. 6a(i) shows the scatter plots of the peak amplitudes versus dwell times of the ion current peaks observed for the translocation of unoccupied carriers. All events which resemble the shape of double peaks are shown in orange, and all other events as black triangles, indicating that they cannot be classified. To eliminate outliers and to ensure robust classification, ion current events will only be classified as a true double peak representing an unoccupied carrier (indicated by orange circles) if the measured peak amplitude and dwell time fall within the 95% confidence ellipse, which is indicated in the figure. Peaks which fall outside of this boundary are indicated by orange triangles and will not be considered as events representing unoccupied carriers.

Similarly, Supplementary Fig. 6a(ii) shows the scatter plots of the peak amplitudes versus dwell times of the ion current events observed for the translocation of carriers incubated with ten times excess of CRP expected to lead to the majority of carriers being occupied. All events which resemble the shape of double peaks and single peaks are shown in orange and blue, respectively, and all other events as black triangles, indicating that they cannot be classified. As above, to ensure a robust classification for single peaks to represent CRP-occupied carriers, the 95% confidence ellipse (indicated as a blue ellipse) is employed as an in-out filter. Only single peaks which fall within this 95% confidence area are considered as resulting from the translocation of a CRP-occupied carrier (indicated by blue circles), all other single peaks are considered unclassified (indicated by blue triangles). To classify the double peaks, and thereby establishing the number of events representing unoccupied carriers, the confidence ellipse from panel (i) is indicated in orange, and now only double-peak events which fall within this area are considered as representing unoccupied carriers (orange circles), while the ones outside this area are dismissed (orange triangles).

This now enables the classification of observed ion current peaks into three categories – double peaks representing unoccupied carriers, single peaks representing CRP-occupied carriers, and unclassified peaks which resemble neither a double nor a single peak. This multi-parameter classification allows the discarding of ambiguous translocation events, for example resulting from broken carriers, in a robust way. Such events would likely resemble single peaks and hence represent false positives. To illustrate this, the DNA nanostructures were deliberately disrupted by incubation in 10 mM $CaCl_2$ for 30 min to substitute the constituent $Mg^{2+}$ with $Ca^{2+}$ prior to translocation. The AFM micrographs in Supplementary Fig. 6b panel (i) demonstrate that the carriers have been degraded significantly. During the translocation experiments only very few events were recorded, and the peak amplitude vs dwell time scatter plot shows that none of the recorded peaks fall within the relevant confidence ellipse (panels (ii) and (iii) of Supplementary Fig. 6b), demonstrating the robustness of the classification approach. As such, the counting of false positives from broken or truncated carriers in the sample is limited effectively by the filtering of the single peaks via the classification procedure discussed above. We note that the small concentration of $Ca^{2+}$ in the translocation buffer does not affect the carriers significantly. Even after incubation of the carriers for 4 h in the translocation buffer, only a small number (~10%) of the translocation events would be classified as CRP-occupied carries with the above classification method (Supplementary Fig. 6c).

Importantly, due to the large dimensions of the nanopipette pore (100 nm) compared to the diameter of CRP (11 nm; Fig. S5), the translocation of CRP alone does not lead to a detectable ion current signature and hence cannot be detected by our nanopipette sensor (Fig. 2d). Similarly, any other molecules in the solution, including linear DNA which have been used as carriers in previous single-molecule sensing[14,32] (Supplementary

Fig. 7a), would also not lead to any signals, thereby neither contributing noise nor false negatives or positives.

**Quantitative single-molecule biosensing.** To demonstrate quantitative sensing using the translocation of carriers with specific DNA aptamers, and the concept of counting individual carrier molecules classified through the three-parameter approach (peak amplitude, dwell time, and shape), we analysed the ion currents of a range of translocation experiments at different CRP concentrations (Supplementary Fig. 7b).

Figure 3a shows a collection of representative ion current peaks for a range of different CRP concentrations. For each concentration, the observed ion current events were classified as described above (Fig. 3b and Supplementary Fig. 7c). Where single or double peaks did not satisfy the filtering criteria they were marked as 'unclassified' and were not taken into account for the concentration analysis. Such unclassified peaks represented between 9 and 36% of the total number of events in a 2-min trace.

Figure 3c shows the normalised single peak count, i.e., classified single peaks vs total number of classified peaks, for different concentrations of CRP from 3 nM to 90 nM. As expected, the normalised single peak count increases with increasing CRP concentration. The data were fitted with a Langmuir isotherm, using the dissociation constant $K_d$ as the only fitting parameter, and the result is shown as a solid line. The $K_d$ obtained from the fit is $11 \pm 2$ nM, which is of the same order of magnitude as the results from our SPR study of DNA aptamer 1.

To investigate the specificity of our sensing system, and in particular of the carriers to the CRP target, a random DNA sequence was selected to act as a non-specific aptamer and the translocation ion current was measured for carrier concentration of 9 nM and CRP concentration of 90 nM, i.e., the highest concentration reported in Fig. 3. The analysis of the ion current events is shown in Fig. S8. A single distribution of peak amplitudes and dwell times with averages of $69 \pm 5$ pA and $0.3 \pm 13$ ms, respectively, were found consistent with the values measured with unoccupied carriers. The scatter plot clearly shows that only double peaks were identified, and no single peaks, demonstrating that a non-specific aptamer does not lead to any detection signal.

Furthermore, the CRP-specific carrier (at a 9-nM concentration) was subjected to 90 nM of a control protein of similar size as CRP (MupB), and the results of the ion current analysis is shown in Figure S8. Similar to the non-specific aptamer, single distributions of dwell time and peak amplitudes (averages of $0.25 \pm 0.1$ ms and $60 \pm 7$ pA) were observed which are in line with those for unoccupied carriers. The scatter plot clearly shows that no single peaks were identified ($n > 50$) demonstrating that no MupB bound to the CRP-carriers.

Using the three-parameter classification (amplitude, dwell time, and ion current signature) quantitative detection down to 3 nM CRP in ~5 μl sample volume was achieved within a 2-min sampling window.

Importantly, a similar study but with a different CRP-specific aptamer (aptamer 2) was carried out and the data is presented in Supplementary Tables 5 and 6. The results are shown in Supplementary Figs. 12 and 13 and very similar results to the carrier version with CRP aptamer 1 were obtained, demonstrating the robustness of the biosensing approach.

For applications in clinical diagnostics, it is important that quantitative detection of analytes such as CRP can also be performed in complex biological fluids. To demonstrate the performance of our sensor system with such fluids, nanopipettes were filled with a solution of 5% human plasma diluted in 0.1 M

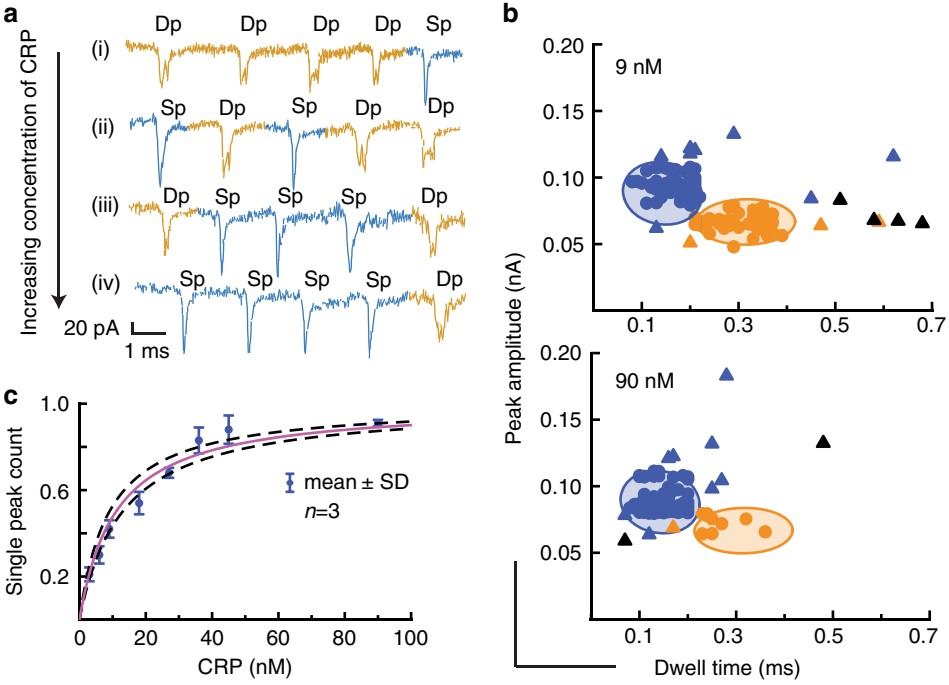

**Fig. 3 CRP binding study. a** Representative selection of ion current signatures for different CRP concentrations. The peak traces are stitched together from individual peaks of a longer trace to remove regions with no events for the purpose of illustration. **b** Translocation events of carriers (9 nM) incubated with different concentrations of CRP with their peak amplitude plotted against dwell time. The single peaks in the ion current data are coloured blue while the double peaks are coloured orange, and the unclassified events discarded from the quantitative analysis are represented as triangles. The plots are overlaid with the 95% confidence ellipses from Fig. S6a. **c** Normalised single peak count, i.e., ratio of single peaks vs total classified peaks against CRP concentration (Supplementary Tables 1 and 2). The data were fitted with a Langmuir isotherm (solid line) and revealed a $K_d$ of 11 ± 2 nM. The dashed lines represent the confidence boundaries of the fit. The error bars denote the standard deviation of translocation experiments conducted on different days using three different nanopipettes ($n = 3$). The sampling time was 2 min for all concentrations. Source data are in the Source Data file.

KCl and spiked with different concentrations of CRP ranging from 3 nM to 36 nM. Similar to the situation in buffer, the carriers in 5% plasma without CRP produced double peak ion current events, but with a slightly shorter dwell time of 0.26 ± 0.10 ms and peak amplitude of 55 ± 6 pA (Fig. 4a, $n > 50$). In contrast, the peak characteristics for occupied carriers, i.e., the single peaks observed in 5% plasma spiked with CRP were found to be consistent between experiments in 5% plasma (peak amplitude 0.12 ± 0.04 ms and dwell time 98 ± 1 pA, $n > 30$, Fig. 4b) and buffer (0.15 ± 0.05 ms and 90 ± 9 pA).

The different ion current events observed during a translocation experiment with 5% plasma can be classified into single peaks, double peaks and unclassified peaks in the same way as in buffer. Figure S9 shows the scatter plots for peak amplitude versus dwell time for all CRP concentrations with the single and double peaks which were selected as representing CRP-occupied and unoccupied carriers, respectively, indicated in the same way as for the buffer experiments.

The normalised single peak count, i.e., the ratio of single peaks vs total classified peaks, is shown in Fig. 4c as a function of CRP concentration. Similar to the results in buffer, the normalised single peak count increases with CRP concentration, and follows a similar behaviour. The limit of detection is estimated to be 9 nM (compared to 3 nM in pure buffer). We note that the number of unclassified events observed for various CRP concentrations in plasma samples were similar (9–25%) to those observed in pure buffer.

## Discussion

We have demonstrated a biosensing approach based on identifying and counting individual biomarkers translocated through a nanopore on a DNA origami carrier which features a central cavity with a target-specific DNA aptamer.

To demonstrate the underlying concept, we have produced three DNA nanostructures of identical outer dimension but featuring central cavities of different sizes by generating customised single-stranded scaffolds for DNA origami folding. We found that upon translocation of these DNA nanostructures through the 100-nm pore of a glass nanopipette, the ion current through the nanopore was modulated differently depending on the size of the central cavity in the DNA origami. The origami with no central cavity caused a single peak in the translocation ion current, while both origami with central cavities led to double peaks, albeit with different double-peak heights. Furthermore, the peak characteristics, the peak amplitude and dwell time, were found to be significantly different depending on the presence or absence, and in fact the size, of a central cavity. Together, the three characteristics – peak shape, amplitude and dwell time – provide a robust way to differentiate between carriers with and without a cavity. This was exploited to generate a biosensor platform for human CRP in buffer and in diluted human plasma.

We designed DNA origami structures with central cavities large enough to lead to a clearly identifiable double-peak in the translocation ion current, but small enough that if the biomarker of interest – human CRP – is inserted into the cavity, the translocation ion current peak becomes a single peak. To facilitate the specific and selective binding of the biomarker of interest into the cavity, a biomarker-specific DNA aptamer was weaved into the DNA origami structure on the edge of the cavity. Translocation studies with unoccupied DNA origami carriers and highly CRP-occupied carriers revealed two distinct dwell-time – peak amplitude clusters, which correlated with single and double peak shapes, respectively.

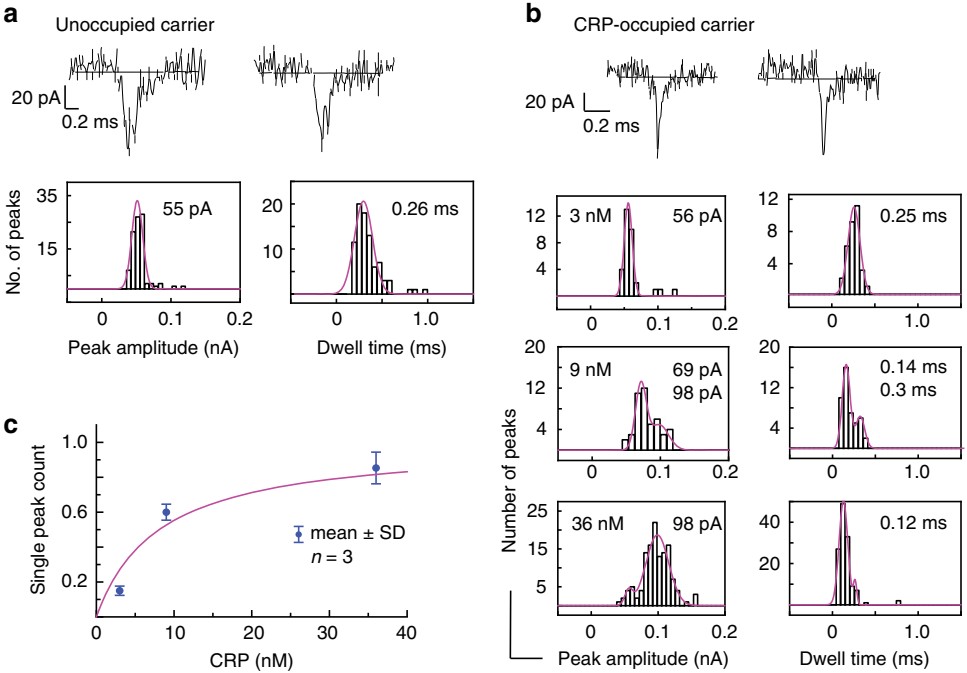

**Fig. 4 CRP detection in diluted human plasma.** Typical ion current traces, and peak amplitude and dwell time histograms for **a** unoccupied carriers and **b** CRP-occupied carriers incubated in different concentrations of CRP in 5% plasma. **c** Normalised single peak count, i.e., ratio of single peaks vs total classified peaks against CRP concentration (Supplementary Tables 3 and 4). The solid line is a guide to the eye. The error bars are standard deviation calculated from three different traces ($n = 3$). The sampling time was 2 min for all samples. Source data are in the Source Data file.

This ability to differentiate between biomarker-occupied and unoccupied carriers provides solid foundations for biosensing.

Using the three-parameter classification, i.e., the peak shape and the 95% confidence ellipse in the peak amplitude – dwell time scatter plot, quantitative detection down to 3 nM and 9 nM CRP in ~5 µl sample volume of buffer and diluted human plasma, respectively, was achieved within a 2-min sampling window.

Using robustly folded rather than long linear DNA which is often the carrier of choice for nanopore experiments, many of the setbacks were overcome. The easily identifiable ion current signature of the DNA origami carrier not only minimises false positives owing to knots and folds often occurring in linear DNA, but also eliminates the 'tail to head' and 'head to tail' ambiguity which is encountered with long linear DNA strands with respect to its translocation direction.

Importantly, this carrier/nanopore biosensing approach is based on counting individual biomarkers rather than relying on an ensemble-averaged signal. As such, the sensitivity of this system can be improved, for example, by reducing the size of the confidence ellipses used to eliminate outliers and thus to increase the stringency of the classification. This, in turn, would require an extension of the sampling time to ensure statistically relevant numbers of occupied and unoccupied carriers are recorded. Furthermore, improving the DNA origami designs with increased robustness will reduce the number of broken and thus unclassifiable carriers in the sample, contributing to increased sensitivity and specificity. Finally, the presented biosensing system has the potential to be advanced to multiplexed detection, e.g., through using ribbons of DNA origamis with integrated barcoding.

## Methods

**DNA origami design**. The three concentric square origamis were designed using the custom made single-stranded DNA (ssDNA) scaffolds of 9073 bp, 8515 bp and 6307 bp whereas the DNA nanostructures used as carriers in this project were designed from 7249 bp m13mp18 ssDNA from Tillibit (Garching, Germany). The procedure for custom made scaffolds is provided in the Supplementary

information. All short staple strands for the respective nanostructure designs were purchased from IDT (Coralville, IA, USA).

All DNA origami nanostructures were designed using the caDNAno 2.3 software (the routing design for each nanostructure is included in the Supplementary information, Supplementary Fig. 11d and Fig. 14) The source data (caDNAno design files and oligo list) is provided along with this paper in the data availability section. For the origami folding, a mixture of ssDNA scaffold with a 10-fold molar excess of the staple strands in the folding buffer containing 10 mM TrisAc (pH 7.4), 10 mM MgAc and 1 mM EDTA (Sigma Aldrich, USA) was heated to 95 °C and then cooled to room temperature with 1 °C/min. The folded structures were purified by removal of excess staples via Sephacryl S400 (GE healthcare, UK) size exclusion column and eluted into the same folding buffer.

The aptamer sequence containing a modified end sequence with 5′ (aptamer 1) and 3′ (aptamer 2) amine attachment was incorporated into the carrier design in a second thermal annealing step (heating up to 35 °C followed by decrease to room temperature at 0.5 °C per minute) either in the presence of the staples before the purification step, or alternatively immediately after the purification step. Either way resulted in precisely folded frame DNA origami carriers inserted with an aptamer.

Specific hCRP aptamer 1:
NH2-AAGCCTTTATTTCAACGGCAGGAAGACAAACACGATGGGGGGG TATGATTTGATGTGGTTGT

TGCATGATCGTGGTCTGTGGTGCTGT

Specific hCRP aptamer 2:
CGAAGGGGATTCGAGGGGTGATTGCGTGCTCCATTTGGTGTTTTTTTT TTTTGCAAGGATAAAAATTT-NH2

Non-specific random aptamer:
NH2-CTGAACAAGAAAAATAGCAGAACTTACGAGCCAGGGGAAACAG TAAGGCCTAATTAGGTAA

AGGAGTAAGTGCTCGAACGCTTCAGA

**SPR study**. Aptamer–protein binding studies were carried out via an ESPRIT SPR instrument from Metrohm Autolab B.V (Utrecht, The Netherlands). For this, the end-modified aptamers attached with an amine group were anchored onto the gold SPR surface functionalized with a self-assembled monolayer (SAM) of $C_{11}PEG_6$-COOH via EDC (1-ethyl-3-(3- dimethylaminopropyl) carbodiimide) NHS (N-hydroxysuccinimide) coupling chemistry[44]. The gold SPR surface from XanTec bioanalytics was cleaned by sonication twice in acetone for 10 min and then immersed in an ethanolic 1 mM SAM solution containing 5% acetic acid and allowed to stand at room temperature for 48 h. Following the incubation, the surface was rinsed with 100% ethanol, quickly dried with nitrogen and mounted on the SPR system. To attach the DNA aptamers to the gold surface, the COOH-terminated SAM surface was activated with 50 mM EDC and 200 mM NHS in 100 mM MES buffer at pH5.5 for around 15 min. The surface was washed with the MES buffer, followed by a 30-min incubation with 5 µM of DNA aptamer in 10 mM sodium acetate buffer at pH5.5.

Any remaining activated COOH sites were quenched by exposing the surface to 100 mM ethanolamine in water for ~10 min. The surface was then washed with the binding buffer (10 mM TrisAc, 10 mM MgAc, 2 mM $CaCl_2$ and 1 mM EDTA in water) and then challenged with varying concentration of human CRP. The binding was measured as the change in resonance angle.

**Nanopipette fabrication and ion current measurements.** The nanopipettes with 100 nm diameters were fabricated from glass capillaries of 0.5 mm inner diameter (QF100-50- 7.5, World precision Instruments, UK) using a Sutter instrument model P-2000 laser puller. The pulling protocol comprised two separate lines with the parameters HEAT 575 FIL 3 VEL 35 DEL 145 PULL 75 and HEAT 900 FIL 2 VEL 15 DEL 128 PULL 150. This protocol produced highly consistent glass nanopipettes with a standard deviation of <±12 nm with nanopipettes pulled on different days. An Ag/AgCl wire (0.25 mm diameter, Sigma Aldrich, UK) was inserted into the nanopipettes as a working electrode.

For the translocation experiments the nanopipettes with the working electrode were filled with the translocation buffer (0.1 M KCl with 10 mM TrisAc, 10 mM MgAc, 2 mM $CaCl_2$ and 1 mM EDTA) containing the DNA origami and analyte where applicable at a final concentration of 500 pM. The Mg is required to maintain the stability of the DNA origami, and the Ca to match the buffer conditions used to select the DNA aptamers employed as binding moieties. The grounded counter electrode was immersed in a 0.1 M KCl solution to complete the circuit. On application of a negative potential to the working electrode inside the nanopipette, DNA origami carriers from inside the nanopipette are translocated out into the electrolyte solution resulting in a modulation of the ion current. Measurements were recorded using the Axopatch 700b amplifier (Molecular devices, USA), and the data were acquired at a rate of 100 kHz and 20 kHz low-pass filtered via the PClamp 10.6 software. Initial data analysis was carried out with a custom MATLAB script (provided by Prof Joshua Edel, Imperial College, London, UK) using MATLAB 9.5 and further data analysis was carried out using proFit 7 (QuanSoft, Switzerland). The raw data files are provided along with this paper in the data availability section.

**AFM.** DNA origami samples were deposited on freshly cleaved mica discs for 10–15 mins at room temperature and topped up with scanning buffer containing 10 mM TrisAc and 10 mM MgAc. For observing protein binding to carriers, 2 mM $CaCl_2$ was included in the scanning buffer similar to the nanopipette translocation experiments. The DNA origami samples were imaged using a Bruker Dimension Fastscan (Santa Barbara, CA, USA) with Fastscan D $Si_3N_4$ cantilevers containing a Si tip in tapping mode in liquid. Images were obtained with scan rates of 20 kHz (256 × 256 pixels) via nanoscope 9.1 and analysed with nanoscope analysis 1.9 software.

**Statistics and reproducibility.** The AFM micrographs shown in the main text Fig. 1b are representative images of successfully folded DNA origami concentric square structures. A typical scan size of 1–2 µ was conducted at least three times for each sample from independent folding experiments. Similarly, Fig. 2b contains representative AFM images of CRP unoccupied and occupied carrier. AFM experiments for carriers incubated with CRP were conducted at least three times each on different days for both aptamer 1 and aptamer 2. Likewise, CRP micrographs in Supplementary Fig. 5 were imaged on two different days, 10 micrographs were collected each day. The gel retardation assay in Supplementary Fig. 4 was conducted for carriers functionalised with aptamer 1 and aptamer 2 on different days using carriers from different batches. Experimental results shown in micrographs of Supplementray Fig. 6b, c were repeated twice on different days. And Supplementary Fig. 7 shows the representative ion current traces of lambda DNA translocation, these translocation experiments were repeated using three different nanopipettes for each concentration (500 pM and 1 nM). Data for all the above experiments is available via the data repository provided along with this paper.

**Reporting summary.** Further information on research design is available in the Nature Research Reporting Summary linked to this article.

## Data availability

Data supporting this work can be freely accessed via the University of Leeds repository: https://doi.org/10.5518/858. Source data is available in the Source Data file. Any other relevant data are available from the authors upon reasonable request. Source data are provided with this paper.

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

## Acknowledgements

The work was in part funded by a grant from the Medical Research Council UK under the grant number MR/N029976/1. M.R. acknowledges the LARS Scholarship by the University of Leeds. A.J.L. acknowledges funding from the University of Leeds. P.A. acknowledges funding from the European Union's Horizon 2020 research and innovation programme under the Marie Skłodowska-Curie grant agreement No. 812398. We thank Prof Joshua Edel (Imperial College London) for sharing a custom-written MatLab script for single-molecule analysis.

## Author contributions

C.W. and P.A. conceived and supervised the research. The nanopipette translocation experiments were conducted and analyzed by M.R.; A.J.L. and M.R. designed and constructed the DNA nanostructures and the SPR studies were conducted by R.S.; M.R. and A.J.L. wrote the paper and all authors edited the manuscript.

## Competing interests

The authors declare no competing interests.
