## [Peer Review File · Nature Communications]

Reviewers' Comments:

Reviewer #1:

Remarks to the Author:

“Rational design of DNA nanostructures for diagnostic: towards single molecule biosensing” by Raveendran et al. is a well written and interesting manuscript that describes the use of 2-d DNA origami to act as a binding site for CRP marker. They use glass nanopipette-based sensing to detect for the presence of CRP-bound origami and report an accurate means of detecting CRP down to ~3 nM.

Given the importance of developing rapid and accurate protocols for detecting CRP (CRP is used in common bloodwork to check for inflammatory-based diseases (i.e. Crohn’s, colitis, lupus, arthritis, etc.)) and the interesting approach that the authors have described for doing so, I am enthusiastic in my recommendation of publication for this manuscript, provided they address my comments below.

1. The title is a little awkward and I also think it undersells this work. “diagnostic” ends abruptly. Also, you are doing single molecule biosensing, I think you’ve moved beyond the “towards”. Also, I would strongly recommend you clearly state that you can detect CRP given its ubiquitous role in present day clinical bloodwork for inflammatory-based diseases. Perhaps something like: “Rational design of DNA nanostructures for single molecule C-reactive protein detection”Feel free to come up with something else...but I think you should think a bit more to maximize your google search hit rate.
2. A general question: Given the 2-D nature of the DNA structure, why do you not find different blockades for when the origami goes through the opening perpendicular to the interface (as shown in the cartoon in Figure 2A) versus parallel?
3. Please include a current and time scale on figures 4A and 4B.
4. In the supplementary section, Fig. S6b (ii) seems to be the only long-time current trace I see that shows the blockades from the DNA nanostructures. Please include some more traces that clearly show that the number of blockade events increases with increasing DNA structure concentration. As it stands now, without the 1-8 number labels in Fig. S6b(ii), I could just as easily argue that you are observing spurious background. In fact, I see two or three upward steps in that same trace that are on the some order of magnitude as the so-called DNA events.
5. Same figure as point 4 above. When you zoom in on the 8 events, why does the #1 event have an equal magnitude blockade as the other events when clearly number 1 in the raw trace appears to be about 3-4 times deeper? Also, you should include a current and time scale on these zoomed in traces.

Reviewer #2:

Remarks to the Author:

The manuscript by Raveendran et al. targets the task of detecting a low concentration of biomarkers in a sample by conjugating highly-charged DNA carriers to the biomarkers, to assist with capture and detection of these complexes using a nanopipette. The electrical signatures from origami carrier are different from protein-loaded origami carrier. The authors demonstrate the detection of C-reactive protein in diluted human plasma, which is impressive since detection in a "dirty" sample using nanopores is typically difficult to accomplish. The authors are well established in single-molecule sensing using nanopores, and the paper was written very well. While the paper is well written, I feel that there are improvements that the authors should make, and note my suggestions below. I have combined minor and major comments:

- In the abstract, I would change "small concentrations" to "low concentrations".
- I have a comment about the way the paper begins: The first sentence of the introduction discusses "Rapid and low-cost detection of disease biomarkers...", but oddly, the references all point out to single-molecule analysis papers, which is highly inconsistent with the claim in the sentence. There are many diagnostics platforms that are currently employed and the state of the art should be introduced to include current practices, followed by an excursion into single-molecule detection. This will avoid confusing readers that biomarker detection is exclusively owned by single-molecule techniques.
- In Figure 1d, it seems that for ConC particles there is no translocation, based on the evidence of a broad dwell-time population at a relatively shallow level. Have the authors considered that ConC does not go through the pore? A voltage dependent study can confirm or reject this hypothesis. Otherwise, it is strange that the same exterior dimensions of a particle as ConA and ConB produces such a broad distribution.
- In figure 2c the dwell time axis has negative value, which should not be shown because it is physically impossible to have a negative dwell time.
- The dynamic range of the measurement is a function of the DNA origami maximum concentration, as the authors show in figure 3c.
- The apparently low number of events shown in figure 3b, as well as figure s6, is a bit concerning. Is it because the pore lifetime is compromised by protein sticking to the pore walls, or because the experiment is intentionally short to demonstrate quick accurate readout? It would be good if the number of overall events for each plot is specified, and perhaps also to mention the overall sampling time for each experiment (for example, $n=xxx$, $T=yyy$ min in each panel where data is shown).
- It is important to include a statement on how the error bars were calculated, particularly for the data shown in Figures 3c and 4c. Is it a measurement-to-measurement error, or just a standard error of a single measurement? How many times was each point measured? Based on the table in the SI it seems that the authors have repeated results with 3 pipettes, which is good, but this should be specified throughout the manuscript where data are shown.
- Overall, it seems that detection of all structures is somewhat convoluted by the bandwidth of the measurement. This is evident by events being close to the ~ 100 μ s timescale limit. The question is, how would sensitivity be affected by the exact pipet size?
- Also, it is confusing to me why origami are pulled out from the pipette, rather than being sucked in from the sample chamber. Is this amenable to the study of serum, and possible serial studies where multiple samples are analyzed?
- Finally, it is interesting that the two peaks are seen for the free origami, and would be great to

study this further, perhaps in a separate publication.

We would like to thank the Reviewers for their comments on our manuscript. We have addressed their comments point by point below and revised the manuscript accordingly.

Reviewer 1

1. *The title is a little awkward and I also think it undersells this work. “diagnostic” ends abruptly. Also, you are doing single molecule biosensing, I think you’ve moved beyond the “towards”. Also, I would strongly recommend you clearly state that you can detect CRP given its ubiquitous role in present day clinical bloodwork for inflammatory-based diseases. Perhaps something like: “Rational design of DNA nanostructures for single molecule C-reactive protein detection”....Feel free to come up with something else...but I think you should think a bit more to maximize your google search hit rate.*

We have carefully considered the Reviewer’s comment regarding the title and have changed it to “Rational design of DNA nanostructures for single molecule biosensing”.

2. *A general question: Given the 2-D nature of the DNA structure, why do you not find different blockades for when the origami goes through the opening perpendicular to the interface (as shown in the cartoon in Figure 2A) versus parallel?*

We cannot determine the orientation of the origami with respect to the interface as it goes through the nanopore. We note, however, that previous work has demonstrated that linear extended DNA molecules align with their long axis along the electric field lines (C. Wälti *et al*, *Appl. Phys. Lett.* 2006, 88, 7–10). It is therefore reasonable to argue that, analogous to the linear DNA, the DNA origami would align with their plane along the electric field lines in a similar way.

The electric field lines pass through the nanopore perpendicular to the interface, and together with the above argument this would suggest that the majority of the DNA nanostructures translocate through the nanopore perpendicular to the interface.

Furthermore, if no preferential alignment existed, we would expect a uniform distribution of all orientations. Clearly, for origamis translocating parallel to the interface the ion current signature would look different than when translocating perpendicular. It is also reasonable to assume that the size of the cavity would matter much less for parallel translocation, and in fact all origami would lead to very similar ion current signatures when translocating parallel, i.e. show single peaks.

As a result, a very significant proportion of events for frame origamis (ConB and ConC) would always be single peaks (say if they translocate with angles $<30^\circ$ to the interface) – this is clearly not the case, suggesting that at least some alignment takes place.

3. *Please include a current and time scale on figures 4A and 4B.*

Figures 4A and 4B has been revised to include the scale bars.

4. *In the supplementary section, Fig. S6b (ii) seems to be the only long-time current trace I see that shows the blockades from the DNA nanostructures. Please include some more traces that clearly show that the number of blockade events increases with increasing DNA structure concentration. As it stands now, without the 1-8 number labels in Fig. S6b(ii), I could just as easily argue that you are observing spurious background. In fact, I see two or three upward steps in that same trace that are on the some order of magnitude as the so-called DNA events.*

The data requested by the Reviewer, which show the increasing number of events with increasing DNA origami carrier concentration, is shown below.

In addition, a few more representative ion current traces for unoccupied and CRP-occupied carriers are shown here with >70 events each for a sampling time of 2 minutes.

We note that entire data set including all raw ion current data sets will be made available via the University of Leeds data repository (a statement to this effect has been added to the manuscript (Data availability)).

5. Same figure as point 4 above. When you zoom in on the 8 events, why does the #1 event have an equal magnitude blockade as the other events when clearly number 1 in the raw trace appears to be about 3-4 times deeper? Also, you should include a current and time scale on these zoomed in traces.

The individual peaks are all scaled to the same height for ease of viewing, but unfortunately, we failed to include the scale bars. We apologize for this oversight and have now corrected the figure, i.e. have included the individual scale bars.

Reviewer 2

1. *In the abstract, I would change "small concentrations" to "low concentrations".*

We have changed "small concentrations" to "low concentrations" in page 1 line 1 of abstract.

2. *I have a comment about the way the paper begins: The first sentence of the introduction discusses "Rapid and low-cost detection of disease biomarkers...", but oddly, the references all point out to single-molecule analysis papers, which is highly inconsistent with the claim in the sentence. There are many diagnostics platforms that are currently employed and the state of the art should be introduced to include current practices, followed by an excursion into single-molecule detection. This will avoid confusing readers that biomarker detection is exclusively owned by single-molecule techniques.*

We agree with the Reviewer and the references in the original manuscript were indeed not cited in the most appropriate places. We have changed the flow of references accordingly as follows.

The first paragraph of the introduction introduces biosensors and diagnostics in general, and now includes references (1 – 4). The second paragraph then introduces the concept of molecule techniques with references (5 – 10) (first sentence of second paragraph). This is then followed by an introduction to nanopores and nanopipettes with references (11 – 16).

3. *In Figure 1d, it seems that for ConC particles there is no translocation, based on the evidence of a broad dwell-time population at a relatively shallow level. Have the authors considered that ConC does not go through the pore? A voltage dependent study can confirm or reject this hypothesis. Otherwise, it is strange that the same exterior dimensions of a particle as ConA and ConB produces such a broad distribution.*

We note that ConC has a significantly larger inner window, making it much more flexible than ConA and ConB, as can be seen in figure 1b. Meni Wanunu's group showed (ACS Nano 2017, 11, 10, 9701-9710) that increased residence time of DNA origami within nanopores can be explained by a physical deformation of the origami during translocation. This is likely to be the case here as well, which in turn leads to the broad dwell time distribution for the ConC structures as seen in figure 1d. This hypothesis is also corroborated by AFM imaging which shows a range of different conformations for ConC, indicative of a less rigid structure when compared to ConA and ConB.

4. *In figure 2c the dwell time axis has negative value, which should not be shown because it is physically impossible to have a negative dwell time.*

Figure 2c has been modified as suggested by the Reviewer.

5. *The dynamic range of the measurement is a function of the DNA origami maximum concentration, as the authors show in figure 3c.*

We agree with the Reviewer, and it would indeed be necessary to vary the DNA origami concentration to move the dynamic range.

6. *The apparently low number of events shown in figure 3b, as well as figure s6, is a bit concerning. Is it because the pore lifetime is compromised by protein sticking to the pore walls, or because the experiment is intentionally short to demonstrate quick accurate readout? It would be good if the number of overall events for each plot is specified, and perhaps also to mention the overall sampling time for each experiment (for example, n=xxx, T=yyy min in each panel where data is shown).*

The number of events for each plots used in figure 3 is provided in supplementary table S2, the sampling time for all the experiments in this work is 2 minutes as stated on page 9 line 2. We have also amended the captions of figures 3 and 4 to specifically include the sampling time and to include a reference to table S2 and S4, respectively.

7. *It is important to include a statement on how the error bars were calculated, particularly for the data shown in Figures 3c and 4c. Is it a measurement-to-measurement error, or just a standard error of a single measurement? How many times was each point measured? Based on the table in the SI it seems that the authors have repeated results with 3 pipettes, which is good, but this should be specified throughout the manuscript where data are shown.*

The error bars in figure 3c are standard deviations of translocation experiments from 3 nanopipettes for each CRP concentration conducted on different days. This is stated in the captions of figure 3 and supplementary table S1. Error bars in figure 4c are standard deviations for 3 sets of ion current traces conducted with one nanopipette for each CRP concentration. This is stated in supplementary table S3. The number of events for each experiment and traces used in figure 3c and 4c is indicated in the supplementary tables S1 and S3 respectively.

We have now amended the caption of figure 4c to clarify this.

8. *Overall, it seems that detection of all structures is somewhat convoluted by the bandwidth of the measurement. This is evident by events being close to the $\sim 100 \mu\text{s}$ timescale limit. The question is, how would sensitivity be affected by the exact pipet size?*

The translocation experiments have all been carried out using different nanopipettes on different days resulting in consistent ion current data. This demonstrates that there is limited dependence on pore size and that it does not significantly affect the peak characteristics.

9. *Also, it is confusing to me why origami are pulled out from the pipette, rather than being sucked in from the sample chamber. Is this amenable to the study of serum, and possible serial studies where multiple samples are analyzed?*

Translocating DNA origami from outside to inside the nanopipette was not possible with our experimental setup and limited sample sizes. The nanopipette setup features a large volume (tens of ml) of electrolyte outside as opposed to a few μl volume inside the nanopipette, making it therefore much more suitable to be used in the configuration as

described here. We agree, however, that an inverse setup could potentially be used as well.

10. Finally, it is interesting that the two peaks are seen for the free origami, and would be great to study this further, perhaps in a separate publication.

We thank the Reviewer for their comment and agree that this would fall outside the scope of the paper.

Reviewers' Comments:

Reviewer #1:

Remarks to the Author:

The authors have addressed my concerns in a satisfactory manner. I recommend publication.

Reviewer #2:

Remarks to the Author:

The authors have addressed well the comments by both reviewers. I believe the manuscript is suitable for publication in its current form.